# Multi-UAV Cooperative Trajectory Planning Based on FDS-ADEA in Complex Environments

**Gang Huang, Min Hu \*, Xueying Yang and Peng Lin**

Department of Aerospace Science and Technology, Space Engineering University, Beijing 101416, China
\* Correspondence: jlhm09@163.com

**Abstract:** Multi-UAV cooperative trajectory planning (MUCTP) refers to the planning of multiple flyable trajectories based on the location of each UAV and mission point in a complex environment. In the planning process, the complex 3D space structure increases the difficulty of solving the trajectory points, and the mutual constraints of the UAV cooperative constraints can degrade the performance of the planning system. Therefore, to improve the efficiency of MUCTP, this study proposes MUCTP based on feasible domain space and adaptive differential evolution algorithm (FDS-ADEA). The method first constructs a three-dimensional feasible domain space to reduce the complexity of the search space structure; then, the constraints of heterogeneous UAVs are linearly weighted and transformed into a new objective function, and the information of the fitness value is shared in accordance with the adaptive method and the code correction method to improve the search efficiency of the algorithm; finally, the trajectories are smoothed to ensure the flyability of the UAVs performing the mission by combining the cubic B-spline curves. Experiments 1, 2, 3, and 4 validate the proposed algorithm. Simulation results verify that FDS-ADEA has fast convergence, high cooperative capability, and more reasonable planned trajectory sets when processing MUCTP.

**Keywords:** multi-UAV cooperative trajectory planning; 3D feasible domain space; adaptive differential evolutionary algorithm; key trajectory points; code correction method





## 1. Introduction

Multi-UAV cooperative trajectory planning (MUCTP) is the planning of multiple safe, reasonable, and collision-free flyable trajectories based on the spatial location of UAVs, mission points, and threat obstacles in a complex planning environment [1,2]. Single UAV trajectory planning hardly accomplishes complex military missions due to its limited maneuverability and payload. MUCTP has the advantage of being more flexible and more efficient in performing missions compared with single UAVs; therefore, it gradually replaces single UAVs in performing complex missions. However, MUCTP also increases the difficulty of the combat system; for example, in the calculation process, the complex mountain type and radar detection area generate substantial no-fly data, and the combat system hardly parses these data in a short time; in the cooperative planning process, the heterogeneous UAVs' own constraints and mutual constraints between multiple UAVs may lead to conflicting execution relationships between each UAV and mission point, eventually leading to mission failure [3,4].

In recent years, to improve the planning efficiency of MUCTP, researchers have proposed many algorithms, which have shown good performance in solving MUCTP problems, and these algorithms are mainly divided into three categories:

(1) Cooperative trajectory planning based on intelligent optimization algorithms

Intelligent optimization algorithms are a probability-based random search evolutionary algorithm, which guides the searching direction by constructing heuristic information, such as MUCTP fitness function [5,6]. In [7], the author proposed MUCTP based on mission requirement constraints and studied a new dynamic particle swarm algorithm and

comprehensive learning particle swarm algorithm, which improve the execution speed of MUCTP. Yi et al. [8] proposed MUCTP based on the improved PF-RRT* algorithm, which uses a dichotomous method to create a new parent node at a node near an obstacle instead of updating the parent node in an existing random tree node, thereby greatly reducing the cost of each UAV trajectory.

(2)　Cooperative trajectory planning based on reinforcement learning

　　Reinforcement learning (RL) is one of the paradigms and methodologies of machine learning, where each UAV obtains observation information from the current environment and takes specific behavior or strategies according to this information. At the same time, the learning system evaluates the quality of the behavior, and if the system evaluates the behavior correctly, it conducts positive rewards and strengthens the behavior training to maximize the benefits [9]. Nie et al. [10] adopted the Q-based learning algorithm to solve the shortest trajectory of each UAV and to calculate the cooperative range and obtained the trajectory group of each UAV by adjusting the selection strategy of each UAV. Liu et al. [11] investigated a closed-loop model using the UAV six-degree-of-freedom nonlinear model to predict flight trajectory and adopted a trajectory mapping network based on deep learning to improve the planning calculation speed and prediction accuracy.

(3)　Cooperative trajectory planning based on spline interpolation algorithm

　　The spline interpolation algorithm draws an approximately smooth curve according to key trajectory points discretely in 3D space. However, this method cannot be used alone when solving MUCTP and must be combined with other algorithms to determine the key trajectory points in advance [12]. Qu et al. [13] proposed UAV trajectory planning based on the RL gray wolf optimization algorithm, which determined the key trajectory points by improving the gray wolf algorithm, and then used the B-spline curve to reverse the control points to ensure that the fitted trajectory passed through all key trajectory points. Zong et al. [14] proposed a trajectory planning framework in a 3D dynamic environment, designed a UAV trajectory predictor using the least squares method, and then used a segmented Bezier curve to represent the generated trajectory.

　　The differential evolution algorithm (DEA) is a population-based evolutionary algorithm that was first proposed by Storn et al. in 1995. This algorithm has stable operation, fast convergence speed, and good parallelism, and it performs well in solving global optimization problems with continuous variables [15–17]. However, MUCTP still suffers from many difficulties, which are listed as follows:

(1)　Compared with the two-dimensional plane, the three-dimensional space contains more information within higher complexity and experiences more difficulty in seeking key trajectory points. Although the DEA can search in 3D complex spaces, the complexity of space structures greatly reduces search efficiency. Therefore, how to simplify the space effectively according to the actual mission and environmental information is one of the challenges in optimizing the trajectory.

(2)　The set of key trajectory points is the population individual of the DEA; the key trajectory points can not only display the optimization direction of the algorithm but also display the search efficiency of the algorithm. MUCTP requires quickly finding a better set of key trajectory points in a short time, which means that the diversity and convergence of population individuals must achieve a good balance. Therefore, how to solve the better key trajectory points quickly and accurately is another difficulty in measuring the performance of the algorithm.

　　The abovementioned difficulties must be addressed. In this work, we propose a cooperative multi-UAV trajectory planning based on a feasible domain space adaptive differential evolutionary algorithm (FDS-ADEA) using a centralized computational approach with fixed-wing UAV as the research object. The main contributions of this study are listed as follows:

(1)  In terms of assigning grid point information: The current space segmentation method mostly adopts the 3D grid method, and each grid point has trajectory point information; however, no effective distinguishing information exists between grid points. Therefore, when constructing the grid point form, in addition to taking the UAV sequence, mission point sequence, and obstacle information as the main features, the corresponding trajectory cost information and coordination information are also added to avoid the blindness of the evolutionary direction in the process of algorithm search.

(2)  In terms of constructing space structure: In consideration of the substantial 3D space information, some researchers find key trajectory points through numerous circular iterations, which seriously reduces the search efficiency. Through substantial experimental verification when the location of each UAV and mission point is determined, the planned trajectory often only appears in a limited area on both sides of the vertical section. In accordance with this idea, a 3D feasible domain space, which only calculates the key trajectory point evaluation indexes in the feasible domain as the key trajectory point set, is constructed in this work.

(3)  In terms of constructing the adaptive algorithm: The diversity and convergence of key trajectory points mainly depends on the mutation strategy of the DEA, and determining the relevant mutation strategy through repeated testing is time consuming. Therefore, an adaptive rule based on fitness values is first constructed, and each individual can choose their own mutation strategies according to the adaptive rule in this work. Second, a reasonable coding correction method is used to delete or supplement eligible individuals. This method breaks away from the limitations and constraints of balancing exploratory and exploitative, increases the feasible domain coverage of the search space, and improves the efficiency of searching key trajectory points.

The rest of this paper is organized as follows: related work about simplifying 3D space structural models is presented in Section 2; Section 3 provides the 3D environment model, multi-UAV cooperative constraints, and objective function; Section 4 describes the proposed algorithm in detail; and in Section 5, the performance of FDS-ADEA is compared with other algorithms. Finally, a summary is presented in Section 6.

## 2. Related Work

First, the common methods to simplify the 3D simulation space are described in this section. Second, the basic concept and principle of DEA are introduced.

### 2.1. Simplify 3D Simulation Space

Compared with the 2D plane, the 3D simulated space structure contains substantial information, making it difficult to select key trajectory points. When the traditional 3D grid method is used for trajectory planning, the amount of calculation is often increased due to insufficient information of each grid point. At the same time, when searching the key trajectory points of the trajectory, numerous loops are used to search the optimal trajectory, but the same amount of information between the grid points easily makes the algorithm fall into local optimum.

Han et al. [18] proposed an optimization UAV indoor trajectory planning algorithm based on grids in a complex environment. It was the first time to adopt the GeoSOT-3D method, which was used to reduce the computational complexity of indoor 3D space modeling through spatial subdivision; the method is commonly used in indoor scenes and complex "dead zone" spaces. Liu et al. [19] proposed a spatial representation algorithm based on weight meshing, which first divides and meshes the simulation space and then encodes all meshes. Second, the latitude, longitude, and height coding of the simulated spatial mesh is used to represent the geometric space, which is often used in the scene of outdoor wide-area space but not dense obstacles. Nicolas et al. [20] proposed trajectory planning based on grid neighborhood algorithm, which is used in combination with the A* algorithm, and the combined algorithm can remarkably reduce the complexity of the computation.

### 2.2. Differential Evolution Algorithm

The DEA is a robust and efficient heuristic evolution algorithm that is highly competitive in solving continuous and multi-constraint optimization problems [21]. The basic idea of the DE algorithm has the following steps: first, the boundary of the solution is determined in accordance with the optimization problem, and new individuals are generated through the mutation operator; second, the next generation of individuals is generated through the crossover operators; and finally, the high-quality individuals are retained by the selection operator, and the inferior individuals are eliminated. A schematic of the DE algorithm is shown in Figure 1.

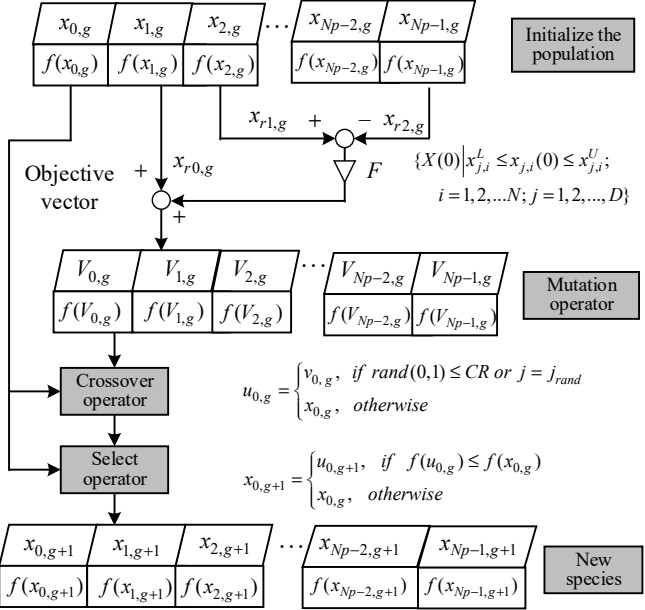

**Figure 1.** Schematic of classical DEA.

(1)    Initialize population

Randomly and uniformly generate $NP$ individuals in solution space:

$$\left\{ X(0) \,\middle|\, x_{j,i}^L \leq x_{j,i}(0) \leq x_{j,i}^U; i = 1, 2, ..., NP; j = 1, 2, ...., D \right\}, \tag{1}$$

$$x_{j,i}(0) = x_{j,i}^L + rand(0,1) \cdot (x_{j,i}^U - x_{j,i}^L). \tag{2}$$

where $NP$ represents the population size; $x_{j,i}^L$ and $x_{j,i}^U$ denote the boundary of the *j*-th "gene" of the *i*-th "individual" in the population, respectively, when the iteration number equal to 0 is the *j*-th gene" of the *i*-th "individual."

(2)    Mutation operator

A mutation operator is one of the key factors affecting the performance of the DE algorithm, and an excellent mutation strategy must have good exploratory and exploitative characteristics. Xu et al. [2] proposed a multi-strategy fusion mutation operator, which can adaptively determine the values of the control parameters $F$ and $CR$. Yu et al. [22] proposed a rank-based mutation strategy, providing individuals with improved fitness values and a higher selection probability to improve the development performance of the algorithm. Three commonly used mutation strategies are listed below:

- DE/rand/1

$$v_{j,i}(g+1) = x_{j,r1}(g) + F \times (x_{j,r2}(g) - x_{j,r3}(g)), \tag{3}$$

- DE/best/1

$$v_{j,i}(g+1) = x_{j,best}(g) + F \times (x_{j,r1}(g) - x_{j,r2}(g)), \tag{4}$$

- DE/current-to-best/1

$$\begin{aligned} v_{j,i}(g+1) &= x_{j,i}(g) + F \times (x_{j,best}(g) - x_{j,i}(g) + \ldots \\ &\quad \ldots + x_{j,r1}(g) - x_{j,r2}(g)) \\ v_{,j,i}(g+1) &= x_{j,i}(g) + F \times (x_{j,best}(g) - x_{j,i}(g) + \ldots \\ &\quad \ldots + x_{j,r1}(g) - x_{j,r2}(g)) \end{aligned}, \tag{5}$$

where the indices *r1*, *r2*, and *r3* are three randomly selected integers; $x_{j,r1}(g)$ represents the *j*-th gene of the *r1* individual in the g-th generation population; $x_{j,best}(g)$ represents the best individual of generation *g*; $v_{j,i}(g+1)$ is the experimental individual; parameter *F* is the scaling factor; and *F* is the scaling factor, which directly affects the global optimization ability of the algorithm.

(3)    Crossover operator

The sequence of offspring individuals is obtained by the variation operator, each of which is selected with a certain probability to generate the test individuals. That is, they are partially swapped from $x_{j,i}(g)$ and $v_{j,i}(g+1)$ to form a new trial vector $u_{j,i}(g+1)$.

$$u_{j,i}(g+1) = \begin{cases} v_{j,i}(g+1), & if \quad rand \leq CR \quad or \quad j = j_{rand} \\ x_{j,i}(g), & otherwise \end{cases}, \tag{6}$$

where *CR* is the crossover rate, which determines the degree to which the number of experimental individuals is exchanged with the parent individual.

(4)    Selection operator

The DEA uses a greedy algorithm to select individuals to enter the next generation of populations based on fitness values, as shown in Equation (7), which calculates the fitness values of the test vector $u_{j,i}(g+1)$ and the original vector $x_{j,i}(g)$, respectively, and selects the better individuals to enter the next generation.

$$x_{j,i}(g+1) = \begin{cases} u_{j,i}(g+1), & if \quad f(u_{j,i}(g+1)) \geq f(x_{j,i}(g)) \\ x_{j,i}(g), & otherwise \end{cases}. \tag{7}$$

## 3. Problem Description

In this section, the 3D environment modelling is first introduced, followed by the objective functions of MUCTP, the UAV constraints and the multi-UAV cooperative constraints.

### 3.1. Environment Modeling

In this paper we use a regular structure of radar obstacles and no-fly zone obstacles and irregular terrain obstacles will increase the difficulty of solving cooperative multi-UAV trajectory planning. During environmental modeling, terrain obstacles and threatening obstructions considerably affect the trajectories. This study uses USGS24kdem to read format USGS 1:24,000 digital elevation map files as a terrain obstacle, which contain real longitude arrays, latitude arrays, and altitude arrays. The threat obstacles consist mainly of radar and no-fly zones, presented by hemispheres and cylinders, respectively, and the MUCTP must avoid these risk areas wherever possible. Terrain obstacles and threatening obstructions in the environment are shown in Figure 2.

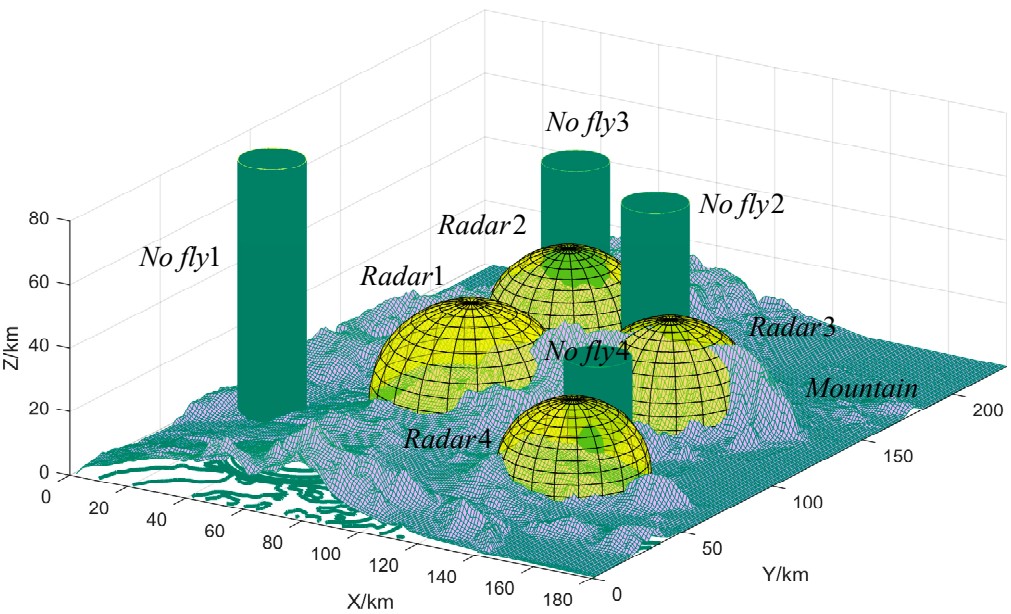

**Figure 2.** Terrain obstacles and threatening obstructions in the environment.

The probability of radar-detecting UAVs is mainly determined by the distance between UAVs and radar. The expression can be expressed by:

$$F_{Radar} = \begin{cases} 0 & d_r \geq R_r \\ (k/d_r)^4 & d_r < R_r \end{cases}, \tag{8}$$

$$d_r = \sqrt{(x_i - x_r)^2 + (y_i - y_r)^2 + (z_i - z_r)^2}, \tag{9}$$

where $F_{radar}$ is the threat value of the radar; $k$ indicates the degree of radar threat; $d_r$ is the distance from the UAV to the center of the radar circle; $(x_r, y_r, z_r)$ is the center of the radar circle; $(x_i, y_i, z_i)$ is the location of the UAV; and $R_r$ is the maximum radius of the radar scan.

The no-fly zone includes air traffic control areas, etc., and it is expressed as follows:

$$F_{zone} = \begin{cases} 0 & d_z \geq R_z \\ R_z^4 / (R_z^4 + d_z^4) & d_z < R_z \end{cases}. \tag{10}$$

where $F_{zone}$ is the threat value of the no-fly zone; $d_z$ is the distance of the UAV from the center of the no-fly zone; $R_z$ is the maximum radius of the no-fly zone.

### 3.2. Establish MUCTP Objective Functions and Constraints

#### 3.2.1. Objective Functions

MUCTP is a set of key trajectory points with optimal trajectory targets in a sequence of key trajectory points that satisfy the constraints. That is, the optimal Pareto solution is found for all missions in the optimization goal. The Pareto solution is a set of improved feasible solutions that are constrained by objective functions. Therefore, the essence of solving the MUCTP problem is to solve the multi-objective optimization problem, while collaborative cooperation is mainly reflected in the cooperation of multiple heterogeneous UAVs to collaboratively complete the specific execution plan planned with a reasonable mission cost and reasonable mission ration.

A new objective function construction method is adopted, and it is mainly considered from three aspects: flight distance cost, execution time cost, and constraint violation cost. The flight distance cost refers to the total flight distance of each UAV to perform a mission. The execution time refers to the maximum time taken by each UAV to complete the mission. The constraint violation cost includes the number of times the key trajectory points cross

obstacles and whether each UAV satisfies the limit constraints. The objective function of MUCTP is described by the following expression:

$$
\begin{aligned}
\min f(x) = {} & \mu \sum_{i=1}^{n} \sum_{j=1}^{m} d_{(i,j)} D_{(i,j)} \\
& + \lambda (\max \sum_{i=1}^{n} \sum_{j=1}^{m} t_{(i,j)} D_{(i,j)}) + \tau \sum_{l=1}^{L} c_l
\end{aligned}
\tag{11}
$$

where $D_{(i,j)}$ is the decision variable between UAV $i$ and mission point $j$; $d_{(i,j)}$ is the flight cost of UAV $i$ to reach mission point $j$; $t_{(i,j)}$ is the time cost of UAV $i$ to reach mission point $j$; $c_l$ is the penalty function corresponding to the constraint, which is set to effectively reduce the number of cases where conflicts occur between UAVs performing mission points. $\mu$, $\lambda$, and $\tau$ are the scale factors of flight distance cost, time cost, and penalty function, respectively, and the three of them are kept on the same magnitude.

Flight distance cost is the most key part of the MUCTP objective function. Flight distance cost can not only show the flight distance of each UAV mission but is also used as one of the indicators to test the distance between key trajectory points and obstacles.

The trajectory of UAV $U$ executing mission point $T$ is taken as an example, as shown in Figure 3. It shows that the 3D trajectory is mapped to the $xoy$ plane. Among them, the dark blue circle and triangle are the UAV departure points $U(x_1, y_1, z_1)$ and location of mission points $T(x_1, y_1, z_1)$, respectively. $A$ represents auxiliary key trajectory points. The dark blue pentagram is a key trajectory point $(k_1, k_2, k_3, k_4)$; the gray semicircle represents the radar scanning area with $r_i$ as the radius; $dist(U, A)$ is the distance between the UAV departure point and the auxiliary key trajectory point $A$; and $dist(k_3, k_4)$ is the distance between key trajectory points $k_3$ and $k_4$. Calculating the distance between trajectory points and obstacles can effectively avoid crossing obstacles.

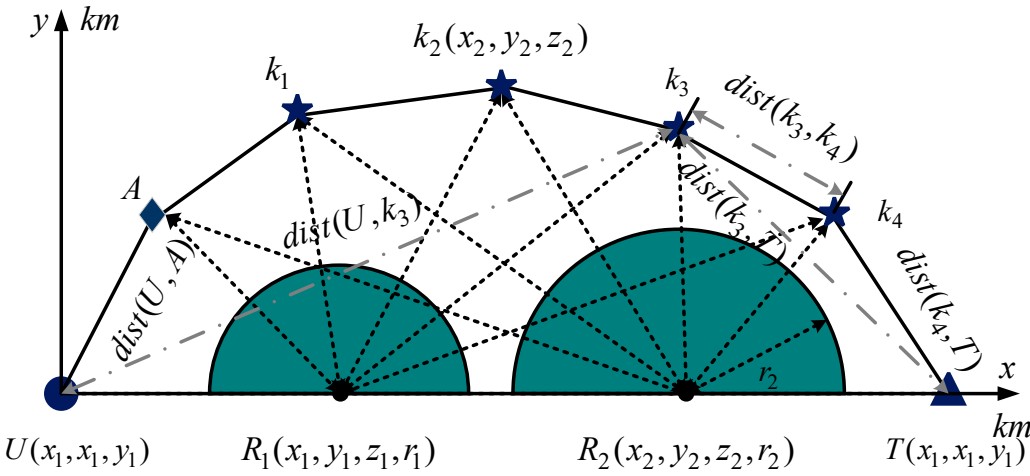

**Figure 3.** Construction of fitness value of objective function.

According to the triangle theorem, calculating the distance between each key trajectory point and the UAV departure point and the mission point can further reduce the height of the key trajectory point. Each UAV flies close to the mountain, which can reduce the probability of being detected by radar. The calculation method of flight distance cost is displayed in Algorithm 1.

---

**Algorithm 1:** The range cost is established in the MUCTP objective function.

---

Input: *Uav* (Unmanned aerial vehicle), *Target* (Mission points), *Auxi* (Auxiliary trajectory points),
*Crtp* (Key trajectory points);
Output: Flight distance cost of key trajectory points;
1 The distance between the auxiliary trajectory point and the departure point is calculated and
assigned to *dist*1;
2 *dist*1 = *sqrt*((*Uav*((1) − *Auxi*(1,1))ˆ2 + (*Uav*(2) − . . .
−*Auxi* (1,2))ˆ2 + *Uav*(3) − *Auxi* (1,3))ˆ2);
3 The distance between each key trajectory point is calculated and assigned to *dist*2;
4  **for** *i* = 1:(size(*Crtp*,1) − 1)
5     *dist*2 = *dist* + *sqrt*((*Crtp*(*i* + 1,3) − *Crtp*(*i*,3))ˆ2 + (*Crtp*(*i* + 1,2) − *Crtp* (*i*,2))ˆ2 + (*Crtp* (*i* + 1,1) −
*Crtp* (*i*,1))ˆ2);
6  **end for**
7 The distance between the mission point and the UAV departure point is calculated and assigned
to *dist*3;
8    *dist*3 = sqrt((*Uav*(1) − *Target*(1))ˆ2 + (*Uav*(2)− . . .
−*Target*(2))ˆ2 + (*Uav*(3) − *Target*(3))ˆ2);
9 Calculate the distance between the key trajectory point and the mission point and the departure
point and record *dis*4;
10   **for** *i* = 1:size(*Auxi*,1)
11     *dis*4 = *dis*4 + (sqrt((*Crtp*(*i*,3) − *Uav*(3))ˆ2 + (*Crtp*(*i*,2) − *Uav*(2))ˆ2 + (*Crtp*(*i*,1) − *Uav*(1))ˆ2) +
*sqrt*((*Auxi*(*i*,3) − *Target*(3))ˆ2 + (*Crtp*(*i*,2) − *Target*(2))ˆ2 + (*Crtp*(*i*,1) − *Target*(1))ˆ2));
12   **end for**
13 Output *dist*1, *dist*2, *dist*3, *dis*4 and calculate the total fitness value;
14 *TotFitness* = 0.15\*dist1 + 0.15\*dist2 + 0.2\*dist3 + 0.5\*dist4;
15 Solu = FDS-ADEA (*TotFitness*); /\* Output key trajectory points \*/
16 **end**

---

### 3.2.2. Constraint Function

Cooperative constraint function and coordination relation:

(1)   UAV and target point decision variable constraints: each UAV must execute a target
point, and each target point must be executed by a UAV.

$$\bigcap_{i=1}^{N} D_{(i,j)} = 1 \ \forall j = 1, ..., M \ \ N = M, \tag{12}$$

where $N$ is the number of drones and $M$ is the number of mission points.

(2)   Constraint of maximum flight distance cost: the total flight distance cost of each UAV
missions point is

$$\sum_{i=1}^{n} \sum_{j=1}^{m} d_{(i,j)} D_{(i,j)} \le \sum_{k=1}^{n} D_k, \forall k = 1, ..., n \qquad , \tag{13}$$

$$C_{1violation} = \begin{cases} 0 & d_{(i,j)} \le D_i \\ 1000 & d_{(i,j)} > D_i \end{cases}, \tag{14}$$

where $d_{(i,j)}$ represents the flight distance cost, and $D_k$ represents the limited maximum
flight distance cost of each UAV. Equation (14) indicates that UAV $i$ is penalized when it
exceeds its limited maximum flight distance.

(3)   Constraints of UAV min/max speed $V(km/h)$

$$V_{(i)} = [V_{(i)min}, V_{(i)max}], \tag{15}$$

$$C_{2violation} = \begin{cases} 0 & V_{(i)min} \le V_{(i)} \le V_{(i)max} \\ 1000 & V_{(i)max} \le V_{(i)} \ \ or \ \ V_{(i)} \le V_{(i)min} \end{cases}, \tag{16}$$

where in Equation (15), $V_{(i)\min}$ is the minimum speed of the UAV $i$, and $V_{(i)\max}$ is the maximum speed of the UAV $i$; Equation (16) indicates that the UAV $i$ exceeds the limited minimum/maximum speed to be penalized.

(4)  Constraint of maximum flight time $T_{\text{time}}$: the maximum time required by each UAV to conduct a mission.

$$\begin{cases} \max\left\{ \sum\limits_{i=1}^{n} \sum\limits_{j=1}^{m} t_{(i,j)} D_{(i,j)} \right\} \leq \sum\limits_{i=1}^{k} T_k & \forall k = 1, ..., n \\ T_{\max} = \max\left\{ \frac{d_{(i,j)} D_{(i,j)}}{v_i}, \forall i, j \in N \right\} \end{cases}, \tag{17}$$

$$C_{3violation} = \begin{cases} 0 & T_{real}(i) \leq T(i) \\ 1000 & T_{real}(i) > T(i) \end{cases}, \tag{18}$$

where, according to Equation (13), maximum flight distance cost and Equation (15) maximum/minimum speed constraint, the maximum flight time can be calculated. Equation (18) means UAV $i$ will be punished for exceeding the limited maximum sailing time.

(5)  UAV maximum payload constraint $U_{missile}$: This constraint reflects the maximum payload of UAVs.

$$\max\left\{ \sum\limits_{i=1}^{n} \sum\limits_{j=1}^{m} U_{missile}(i,j) D_{(i,j)} \right\} \leq U_k \qquad \forall k = 1, ..., n, \tag{19}$$

$$C_{4violation} = \begin{cases} 0 & U_{missile}(i) \leq U_{\max}(i) \\ 1000 & U_{missile}(i) > U_{\max}(i) \end{cases}. \tag{20}$$

(6)  Constraint of time sequence between mission points $T_{sort}$: this constraint reflects the importance of mission points. Important mission points are executed first, and other mission points are executed in accordance with their constraints.

$$\begin{cases} T_j\max < T_{(j+\tau)}\min \\ \forall \tau \in N \quad \text{and} \quad \tau < m - j \end{cases}, \tag{21}$$

$$C_{5violation} = \begin{cases} 0 & T_j\max < T_{(j+\tau)}\min \\ 1000 & T_j\max \geq T_{(j+\tau)}\min \end{cases}. \tag{22}$$

Equation (21) indicates that mission point $j$ must be executed later than mission point $j + \tau$, and $\tau$ is a positive integer. Equation (22) indicates that the mission point is not hit according to the set timing.

(7)  Constraint of wait time $T_{\text{wait}}$: to ensure that UAVs reach their designated targets, some UAVs are allowed to wait for a period of time before leaving.

$$T_{wait}(i) \leq T_{\max}(i), \tag{23}$$

$$C_{6violation} = \begin{cases} 0 & T_{wait}(i) \leq T_{\max}(i) \\ 1000 & T_{wait}(i) > T_{\max}(i) \end{cases}. \tag{24}$$

## 4. Design of FDS-ADEA Algorithm

At present, research on MUCTP simulation space is mostly on 2D simulation space or 3D simulation space and order UAV trajectory planning. The reasons are as follows: (1) The amount of 3D spatial information is large, and key trajectory points are difficult to solve. Although key trajectory points can be found through numerous loop iterations, this method seriously reduces the search efficiency [23,24]. (2) Many algorithms adopt fixed parameter combinations and single-preference mutation strategies, which cannot satisfy the various needs in the dynamic optimization process of MUCTP.

Therefore, this section proposes a multi-UAV trajectory planning based on FDS-ADEA, which constructs a reasonable UAV feasible domain space through the location of UAV, mission points, radar scanning area, no-fly zones, and mountain terrain in space. Then, the adaptive DEA is used to construct the mutation strategy archive and the coding correction method to obtain the key trajectory point set. Finally, the flight trajectory is fitted by the B-spline function. The algorithm framework of FDS-ADEA is shown in Algorithm 2:

---

**Algorithm 2:** Framework of the FDS-ADEA

---

Input: *obj* (The start points of the UAVs: $U_s$; The target points of the UAVs: $T_s$; The population size: *Pop*; The generations: *Gen*); The random sequence population: *Mx*; Auxiliary trajectory points: *Auxi*; Key trajectory points: *k*; Mutation rate: *Pc*; Crossover rate: *Sc*.
Output: Set of key trajectory points:*M t*; Fitted trajectory: *Ft*.
1 Construct threat barriers *Obstacle* according to Equations (8)–(10). Choose a simulated *Terrain*. Determine the coordinates of each UAV $U_s$ and mission point $T_s$ in 3D space.
2 The feasible domain space of each UAV is established to obtain the location of key trajectories;
　　*V* = DirectionAngle (*obj*)
3 The arccosine function is used to define the angle between the key trajectory points of each candidate;
　　*Angle* = TargetArccosine (*obj*, *Auxi*, *k*);
4 In accordance with Equation (11) and Equations (12)–(21), the objective function of MUCTP is established;
　　*fitness* = DEFitness (*obj*);
5 /* The FDS-ADEA algorithm is used to obtain the key flight path points */
　　*Solu* = FDS-ADEAoperate (*obj*, *Mx*);
　　　*for gen* = 1: *obj.Gen*
　　　　*Vs* = MutationOperator (*Mx*); /* *Mutation operator* */
　　　　*Cs* = CrossOperator (*Vs*); /* *Crossover operator* */
　　　　*Ft* = DEFitness (*Cs*); /* *Selection operator* */
　　　*End*
　　*Mt* = Fitnessbest; /*Obtain the best key flight path points */
6 B-spline was used to fit the flight path;
　　*Ft*= SplineFittingFunction (*obj*,*Mt*);
7 Obtain the parameters of each trajectory.

---

### 4.1. Construct Feasible Domain Space for UAVs

Figure 4 is a schematic of setting the feasible region between the starting point and the mission point, where the black circle is the position of the UAV $U(x_i, y_i, z_i)$, the black triangle is the location of the mission point $T(x_j, y_j, z_j)$, and the cuboid represents the feasible domain space for generating key trajectory points. When the execution relationship between UAV $U$ and mission point $T$ is determined, the vertical section between the two is established and expanded into a feasible domain space. Then, the feasible domain space produces key trajectory points $k_n$.

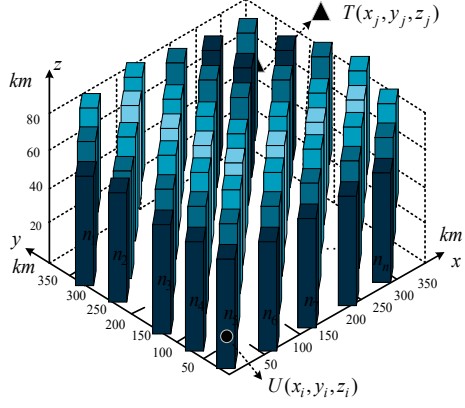

**Figure 4.** Setting the feasible region between the starting point and the mission point.

Figure 5 shows the spatial structure of the feasible domain. The dark blue five-pointed star is the key trajectory point $k = [k_1, k_2, k_3, k_4, k_5]$, the gray dotted line is the execution relationship between the UAV and the mission point, and the hemispherical sphere is the radar scanning area. $V = [xi + \Delta x, yi + \Delta y, zi + \Delta z,]$ is the feasible domain space of the UAV, and each key point is generated in $V$. At the same time, in accordance with the current maneuvering performance of fixed wing UAVs, $120° \leq \alpha \leq 180°$ angles must be satisfied between key trajectory points. This angle range effectively prevents excessive bending of the trajectory point connection. The specific algorithm is shown in Algorithm 3.

---

**Algorithm 3:** Set the feasible flight area of the UAVs

---

Input: $U_s$ (UAV), $T_s$(Mission point), *Radar* (Radar position), $N_k$ (Number of key trajectories), $\Delta x$, $\Delta y$, $\Delta z$ (Range of UAV feasible domains), *Pop* (Population size);
Output: Key flight trajectory points that satisfy the constraints;
1 Construct auxiliary flight path points;
(*xf, yf, zf*) = (*obj.$U_s$*(1) + 10, *obj.$U_s$*(1) + 5, *obj.$U_s$*(1) + 5);
2 Set feasible domain space parameters $\Delta x$, $\Delta y$, $\Delta z$;
3 **while** (*i* < *obj.Pop*)
4 　**while** (*i* < $N_k$)
5 　　*comb* = 1;
6 　　**while** *comb*
7 　　　*C* = [*Start*(*i*, 1) + *rand*(1)*$\Delta x$, *Start*(*i*, 2) + *rand*(1)*$\Delta y$, *Start*(*i*, 3) + *rand*(1)*$\Delta z$];
8 　　　**if** *C* (1,3) > 50; /* *Generate the flight assist plane* */
9 　　　　*C* (1,3) = 50;
10 　　　**end if**
11 *Radar* = (*xi, yi, zi, ri*); /**Extract the threat obstacle parameters* */
12 *Disence* = sqrt(sum (bsxfun (@minus), *Radar,C*). ^2, 2));
/* *Calculate the distance between flight path points and obstacles* */
13 　comb = *any* (*Disence* < *Radius*);
14 　　**end while**
15 The arc cosine formula is used to solve the position of the next flight path point;
16 　　**if** ($120^o \leq \alpha \leq 180^o$) /* *Determine flight path point generation Angle* */
17 　　　*Pop(i + 1,:)* = *C*; /* *is used to judge whether the next point generated satisfies the angle* */
18 　　　*i* = *i* + 1;
19 　　**end if**
20 　**end while**
21 **end while**
22 **end**

---

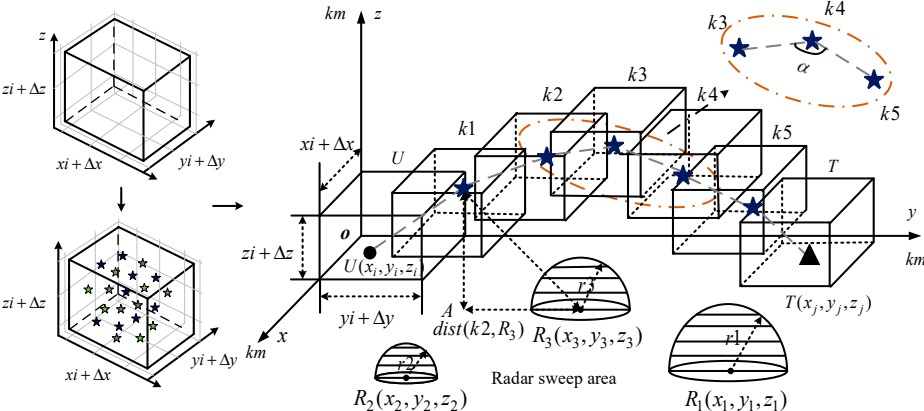

**Figure 5.** Setting the feasible region between the starting point and the target point.

## 4.2. Adaptive DEA

### 4.2.1. Classical Mutation Strategy

Mutation strategy is one of the key factors affecting the performance of the DEA; an excellent mutation strategy must have good exploratory and exploitation capabilities at the

same time and must also maintain a good balance between the two. Currently, the mutation strategies applied to MUCTP are relatively homogeneous, for example, mutation strategies that facilitate solving more key trajectories in the search space: DE/rand/1 and mutation strategies conducive to finding the optimal key trajectory points quickly: DE/best/1. The schematic of the two mutation strategies are shown in Figure 6a,b, respectively.

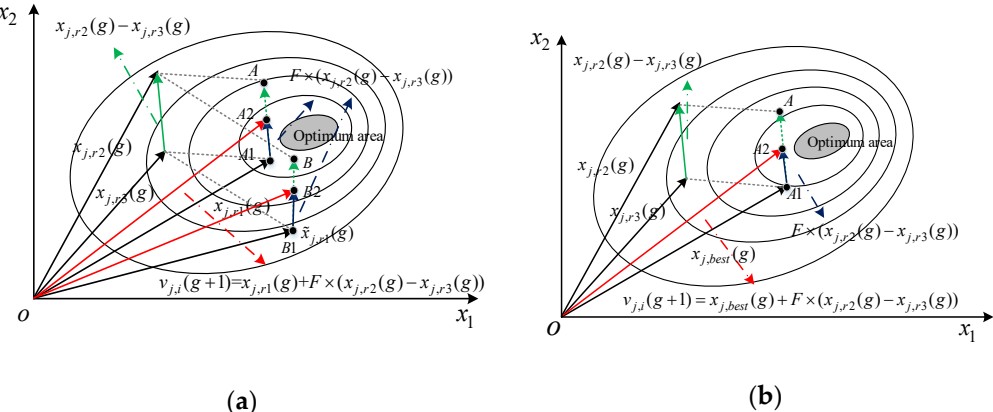

**(a)**  **(b)**

**Figure 6.** Classical mutation strategy. (**a**) DE/rand/1, (**b**) DE/best/1.

In Figure 6a, $\overrightarrow{A1A}$ is the difference vector between individual $x_{j,r2}(g)$ and $x_{j,r3}(g)$, $\overrightarrow{A1A2}$ is the vector after scaling factor scaling, and $o\overrightarrow{A}2$ is the mutation vector obtained by summing the scaling vector with the individual $x_{j,r1}(g)$. The mutation vector $o\overrightarrow{B}2$ can be obtained in the same manner. As shown in Figure 6a, the mutation strategy can effectively expand the exploratory nature and obtain more solutions that satisfy the conditions in the search space.

In Figure 6b, $v_{j,i}(g+1)$ represents the experimental individual obtained by the exploitation mutation strategy; given that individuals with better fitness values are selected during each mutation process, the algorithm is prone to fall into local optimum, and the diversity of individual populations is missing.

### 4.2.2. Adaptive DEA Based on Fitness Values

In recent decades, researchers have designed various adaptive parameter combinations and improved mutation strategies, which show different preferences and good characteristics. However, similar to many practical applications, MUCTP is a "black box" problem. In contrast to its internal structure and characteristics, whether each UAV can effectively complete the corresponding mission must be considered more. Therefore, an adaptive DEA based on fitness values is proposed in this section. A schematic of the algorithm is shown in Figure 7.

In ADEA: First, the entire population is randomly divided into multiple groups with the same number of individuals, and the fitness values of each parent population are calculated and preliminarily classified. Second, the archive of mutation strategies is constructed to store three mutation strategies with different characteristics: (1) mutation strategy I: "DE/rand/1" is beneficial to improve individual exploration, (2) mutation strategy II: "DE/target to best/2" is conducive to balancing the exploratory and exploitation of the population, and (3) mutation strategy III: "DE/best/1" is conducive to enhancing individual exploitation.

Table 1 shows the parameters of *F* and *CR* for each mutation strategy. The better offspring population is composed of better adapted parent individuals, and its main advantage is to guide the population to search for the optimal region quickly. While the primary mission of the inferior population is to maintain diversity, the better or moderate individuals can be regarded as their parent population individuals; therefore, inferior

individuals have access to a great deal of exploitation and exploratory information. The adaptive mutation strategy I, II, and III selection methods are shown in Algorithm 4.

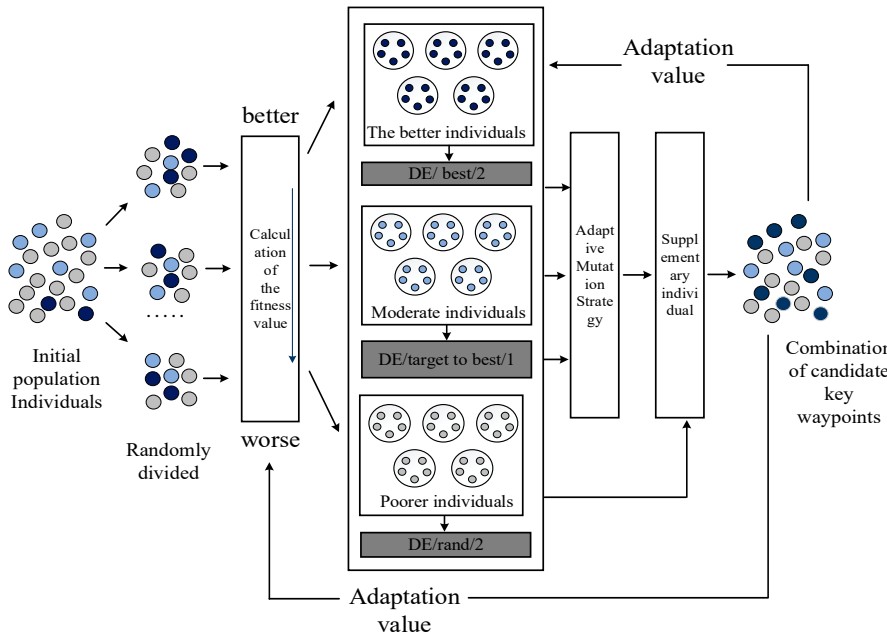

**Figure 7.** Adaptive DEA.

**Table 1.** Parameter selection of mutation strategy.

| Type | Mutation Strategy | Scale Factor $F$ | Crossover Rate $CR$ |
|---|---|---|---|
| Type I | DE/rand/1 | $F = 1.0$ | $CR = 0.9$ |
| Type II | DE/target to best/2 | $F = 0.8$ | $CR = 0.5$ |
| Type III | DE/best/1 | $F = 0.5$ | $CR = 0.1$ |

---

**Algorithm 4:** Adaptive mutation strategy selection

---

Input: $Np$ (Number of individuals); Mutation strategies I, II, III;
Output: A population individual conforming to a fitness value;
1 Determine the population size and randomly divide the population individually;
2 The fitness value of each parent is calculated, and the individual is divided;
3 **for** $i = 1{:}Np$
4   **if** $x_{j,r1}(g)$ is the optimal fitness value, it is the better individual.
5     Mutation strategy III can be selected to produce offspring $v_{j,i}(g+1)$;
$v_{j,i}(g+1) = x_{j,best}(g) + F \times (x_{j,r2}(g) - x_{j,r3}(g))$
6    **else if** $x_{j,r1}(g)$ is the middle fitness value, it is the moderate individual;
7     Mutation strategy III or II can be chosen to produce offspring $v_{j,i}(g+1)$;
$v_{j,i}(g+1) = x_{j,i}(g) + F \times (x_{j,best}(g) - x_{j,i}(g)) + ...$
$\qquad\qquad\qquad + F \times (x_{j,r2}(g) - x_{j,r3}(g))$
8    **else if** $x_{j,r1}(g)$ is the poor fitness value, it is the inferior individual;
9     Mutation strategies III, II, or I can be selected to produce offspring $v_{j,i}(g+1)$;
$v_{j,i}(g+1) = x_{j,r1}(g) + F \times (x_{j,r2}(g) - x_{j,r3}(g))$
10   **end if**
11 **end for**
12 **end**

---

The key trajectory points of MUCTP candidates are all integer discrete sequences. If the sequence values are rounded to the mutation result, then the mutated solution obtains invalid values. If the invalid value continues to participate in the evolutionary calculation, the algorithm stalls. Therefore, invalid discrete values must be corrected, and the coding

correction method can map the invalid sequence directly to obtain a new discrete sequence. For alternative key trajectory points to satisfy planning requirements, the coding correction method must satisfy the following conditions:

Rule 1: Mapping uniqueness: During the correction process, the mapped discrete sequences must be unique. As shown in Figure 8, the invalid sequence individuals are 0, 9, 5. When differential individual 1 is selected as the gene mapping of a newly added individual, the position of the sequence in which the original differential individual 1 is located must be deleted, and the other differential values are mapped in turn. This method guarantees that the mapping is unique.

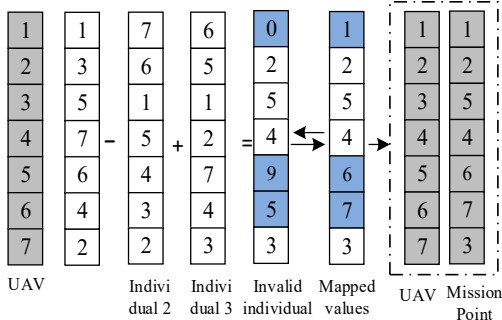

**Figure 8.** Code correction rule.

Rule 2: Mapping validity: The mapped invalid discrete sequences no longer participate in the later variation strategy calculation. Although the mapped sequence is overwritten in Rule 1, the variation strategy information is still stored in the individual. Therefore, the invalid mapping matrix must be removed. At the same time, the mapped individual sequence participates in the later mutation strategy calculation, and this method can improve the performability of the algorithm.

### 4.3. Cubic Spline Curves

The FDS-ADEA algorithm can quickly solve trajectories that satisfy the cooperating conditions, but these trajectories consist of a series of key trajectory point segments, which are not suitable as actual trajectories for UAVs to fly. Therefore, the currently obtained trajectory connection must be smoothed. Yakimenko, O [25] proposed a direct method for rapid prototyping of near-optimal vehicle trajectories. Kaminer, I et al. [26] proposed a cooperative control algorithm for small UAVs, whose main features include trajectory generation for multiple UAVs, which takes into account their aerodynamic characteristics and ensures deconfliction.

In this work, the three B-splines are used to smooth the trajectory, and all the points in the trajectory $L_{ij} = (U_i, Auxi_i, k_{i1}, k_{i2}, ..., k_{ik}, T_j)$ are used as the control points of the B-spline curve. By iteratively curve fitting in 3D space, smooth flyable trajectories can be obtained directly.

When $n = 3$, the basis function of the B-spline can be expressed as:

$$
\begin{cases}
F_{(0,3)}(t) = 1/6(-t^3 + 3t^2 - 3t + 1) \\
F_{(1,3)}(t) = 1/6(3t^3 - 6t^2 + 4) \\
F_{(2,3)}(t) = 1/6(-3t^3 + 3t^2 + 3t + 1) \\
F_{(3,3)}(t) = 1/6(t^3)
\end{cases}
\qquad t = 0, 1, 2, ..., n. \qquad (25)
$$

The cubic B-spline matrix is as follows:

$$
P_{0,3}(t) = \frac{1}{6}[1, \ t, \ t^2, \ t^3]
\begin{bmatrix}
1 & 4 & 1 & 0 \\
-3 & 0 & 3 & 0 \\
3 & -6 & 3 & 0 \\
-1 & 3 & -3 & 1
\end{bmatrix}
\begin{bmatrix}
P_0 \\ P_1 \\ P_2 \\ P_3
\end{bmatrix},
\qquad t = 0, 1, ..., n. \qquad (26)
$$

From the matrix Equation (26), the discrete starting point, auxiliary trajectory point, key trajectory points and mission points of each UAV can be used as the control points of the spline curve to approximate the smooth trajectories of the fitted UAVs.

### 4.4. Flowchart Based on FDS-ADEA Algorithm

Figure 9 shows the flowchart based on FDS-ADEA algorithm, including the construction of the UAV's own constraints and co-constraints, the construction of the MUCTP objective function, and the principle of the adaptive differential evolution algorithm.

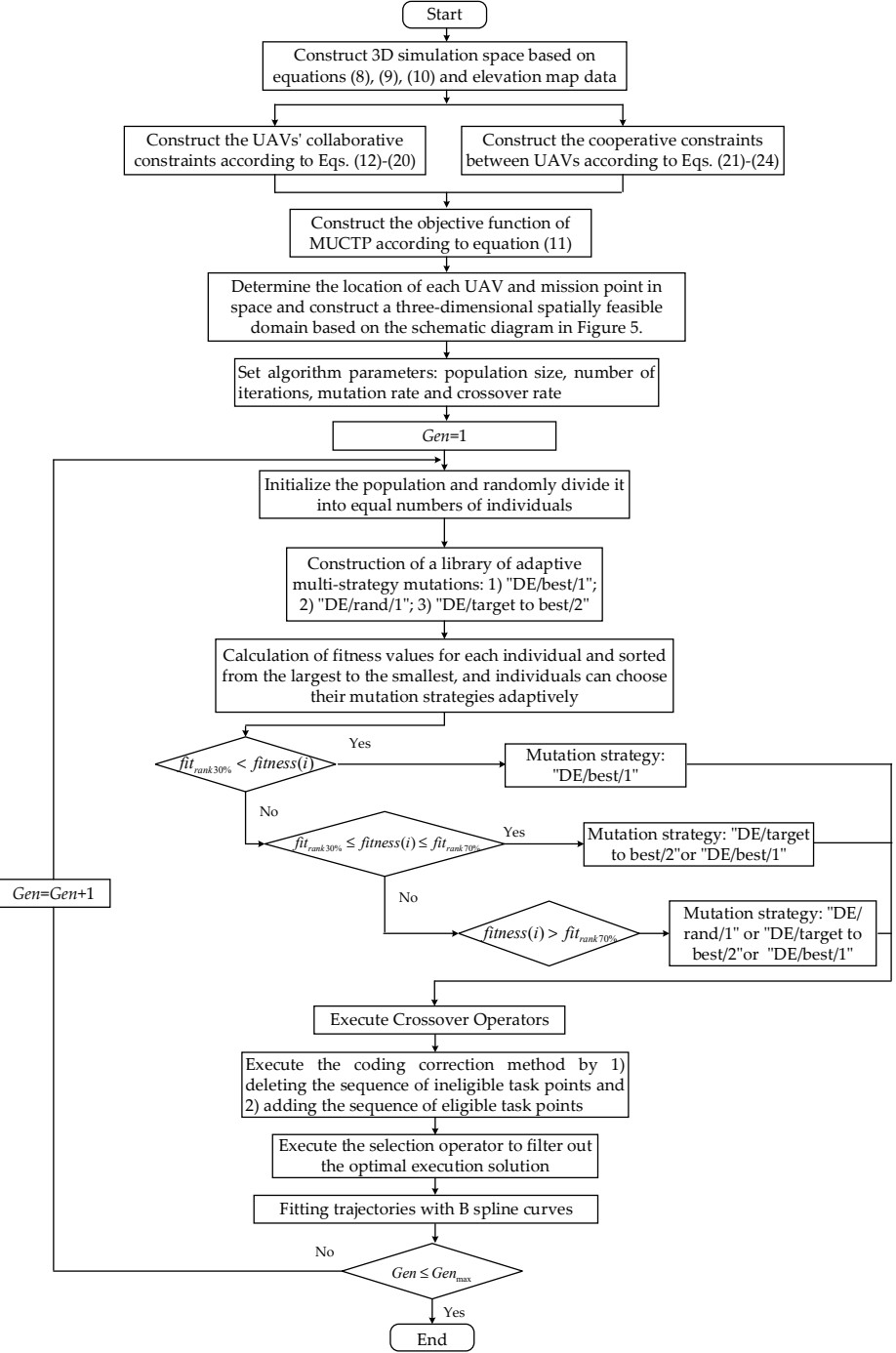

**Figure 9.** Flowchart based on FDS-ADEA algorithm.

## 5. Experimental Simulation, Result, and Analysis

To verify the effectiveness of FDS-ADEA-based MUCTP, this study conducts three sets of simulation experiments in MATLABR2016a. In Experiment 1, single UAV and single mission point are selected to analyze the effectiveness and stability of FDS-ADEA; in Experiment 2, multiple UAV and multiple mission points are selected to analyze the effectiveness of FDS-ADEA; in Experiment 3, the effect of the spatially feasible domain on the performance of FDS-ADEA is analyzed; in Experiment 4, the performance of FDS-ADEA is analyzed in comparison with similar algorithms.

*5.1. Experiment 1: Analyze the Feasibility and Stability of Single UAV Trajectory Planning Based on FDS-ADEA*

Table 2 shows the position of UAV $U_1$ and mission point $T_1$ in the simulation space, respectively. $D_r$ denotes the radar coordinates; $R_r$ represents the scanning radius; $D_z$ and $D_h$ indicate the coordinates and height of the no-fly zone, respectively; $P_n$ is the population size; *Gen* is the iteration number; $P_c$ is the mutation rate; and $S_c$ is the crossover rate.

**Table 2.** Experiment 1: UAV, mission point, environment modeling, and algorithm parameters.

| 0 | U (km) | T (km) | $D_r$ (km) | $R_r$ (km) | $D_z$ (km) | $D_h$ (km) | $P_n$ | Gen | $P_c$ | $S_c$ |
|---|--------|--------|-----------|-----------|-----------|-----------|-------|-----|-------|-------|
| 1 |          |              | (60,180,5)  | 25 | (20,80)   | 80 | 50 | 500 | 0.4 | 0.5 |
| 2 | (0 0 0)  | (140 180 30) | (70,110,5)  | 30 | (50,200)  | 50 | 50 | 500 | 0.4 | 0.5 |
| 3 |          |              | (130,70,4)  | 23 | (100,50)  | 50 | 50 | 500 | 0.4 | 0.5 |
| 4 |          |              | (120,140,4) | 26 | (120,100) | 30 | 50 | 500 | 0.4 | 0.5 |

The verification of the feasibility of the feasible domain space construction is shown in Figure 10; the light-green semicircle is the scanning area of the radar, the dark-green cylinder is the no-fly zone, the yellow circle is the position of the UAV, the yellow pentagram is the position of the mission point, and the red circle is the location of the key trajectory point. The connection of each key trajectory point indicates the correspondence between the UAV and mission point.

As shown in Figure 10a, a total of 12 key correspondences are generated between the UAV and mission point; each correspondence generates 20 sets of feasible domain spaces, and each group has a total of 20 key trajectory points. Figure 10b,c observed the fit between the key trajectory points and the simulation environment from different angles. Figure 10d hides obstacles, such as mountains and radar, to show whether UAV angle constraints are met between trajectory points. The simulation results show that the feasible domain space after planning is relatively uniform, and the maneuverability performance of UAV is in line with each trajectory point. Therefore, the UAV trajectory planning based on FDS-ADEA has good feasibility.

The stability of the feasible domain space construction is verified in Figure 11 and Table 3. *Re* is the correspondence between UAV execution mission points. Considering that individuals have a certain randomness in the search process of intelligent algorithms, the average value of repeated experiments is more representative when the parameters are the same. To verify the stability of the algorithm, four evaluation indexes are selected in this section to evaluate the algorithm performance:

(1)    Average time: $A_t = \sum\limits_{i=1}^{n} Time(i)/Num.$

(2)    Average cost: $A_c = \sum\limits_{i=1}^{n} Cost(i)/Num.$

(3)    Optimal cost: minimum cost of flight in experiments.

(4)    Optimal solution. The optimal solution represents the percentage of the number of times of falling into the local optimal and the number of times currently superior to the average cost in the total number of experiments. The following conclusions can be drawn by combining Figures 10 and 11 and Table 3:

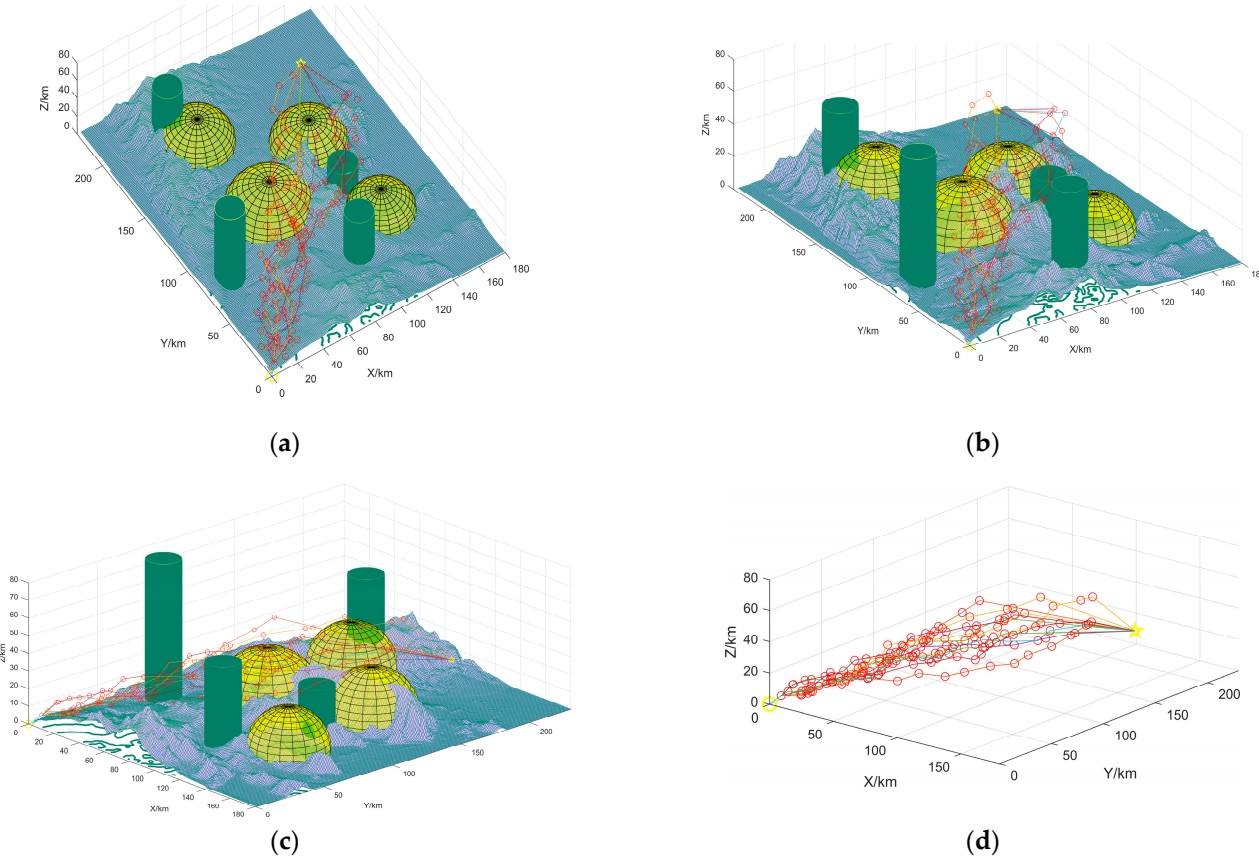

(**a**)

(**b**)

(**c**)

(**d**)

**Figure 10.** Verification of the feasibility of the feasible domain space construction. (**a**) indicates that a single UAV, a single mission point, was selected to verify the feasibility of the feasible domain space with a defined execution relationship; (**b**,**c**) indicate that the feasible domain flight space was observed from different angles, respectively; (**d**) indicates that obstacles were hidden for better observation.

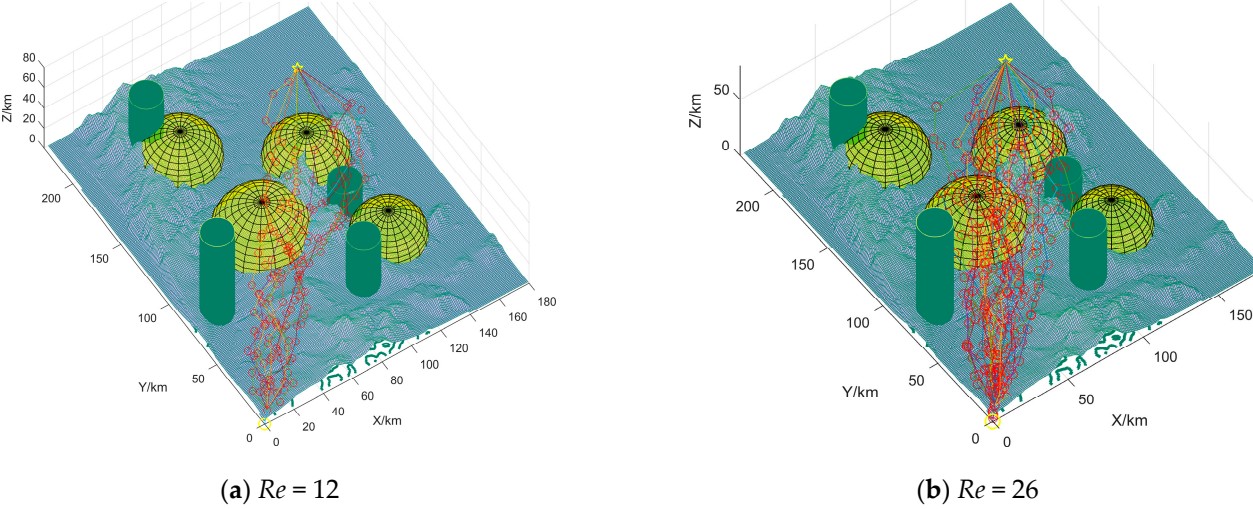

(**a**) *Re* = 12

(**b**) *Re* = 26

**Figure 11.** *Cont*.

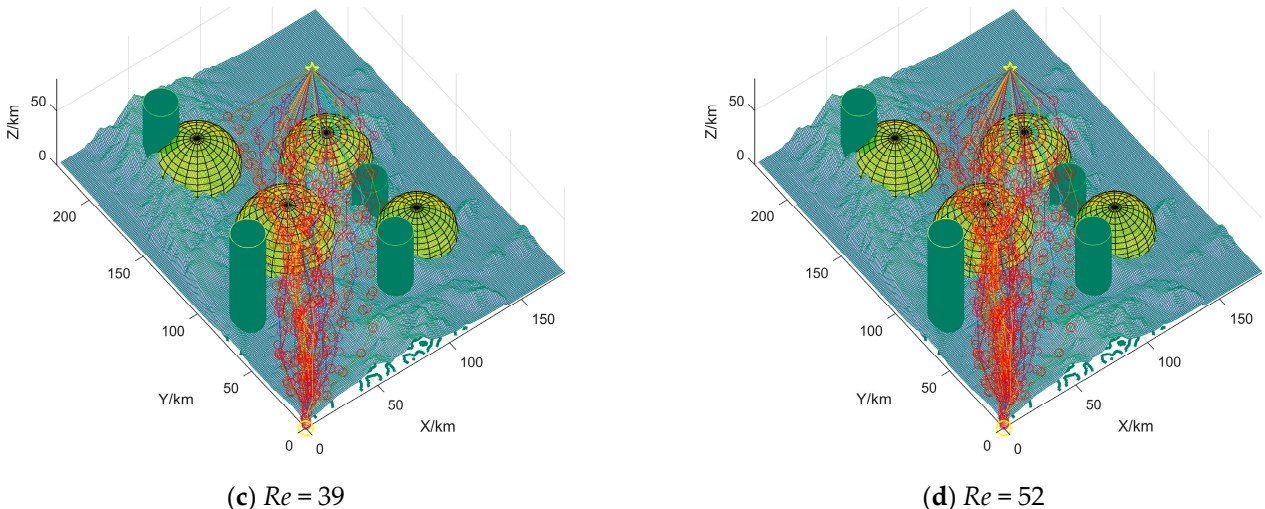

**Figure 11.** Verification of the stability of feasible domain space construction.

**Table 3.** Stability data analysis.

| Type | Correspondence Number | Repetition Number | Average Time (s) | Average Cost (km) | Optimal Cost (km) | Maximum Cost (km) | Optimal Solution |
|---|---|---|---|---|---|---|---|
| | *Re* = 12 | 20 | 16.29 | 603.49 | 600.92 | 635.49 | 91% |
| Experiment 2 | *Re* = 26 | 20 | 28.37 | 606.63 | 603.12 | 637.01 | 85% |
| | *Re* = 39 | 20 | 40.95 | 607.98 | 602.93 | 640.18 | 82% |
| | *Re* = 52 | 20 | 55.22 | 611.32 | 606.10 | 846.06 | 76% |

Conclusion 1: As shown in Figure 10 and Table 3, when the relationship between UAV and mission point execution is determined, the FDS-ADEA-based MUCTP can quickly generate key trajectory points in a short time. The high feasibility, high optimization rate, and short running time indicate that the algorithm has improved exploration and exploitation.

Conclusion 2: As shown in Figure 11 and Table 3, although the number of execution relationships between UAVs and mission points is increased, the average generation value difference between *Re* = 12, *Re* = 26, *Re* = 39, and *Re* = 52 is small, and the optimal generation value is close to the average generation value, which indicates that the trajectory points solved by the feasible domain space of the algorithm are close to the optimal solution.

### 5.2. Experiment 2: Analyze the Feasibility of MUCTP Based on FDS-ADEA

Eight UAVs and eight mission points were selected in Experiment 2. Each UAV was required to strike a mission point, and each mission point was executed by a UAV. Parameters are shown in Table 4. $U_{pos}$, $T_{pos}$, and $R_{pos}$ respectively represent the position of UAV, mission point, and obstacle in space. $D(km)$ is the maximum range of each UAV, $V(km/h)$ is the speed range of each UAV, $T_{time}$ is the maximum flight time of each UAV, $U_{missile}$ is the maximum payload capacity of each UAV, $OrdT$ is the execution importance of the mission point, and $T_{wait}$ is the execution importance of the mission point.

Figure 12d is the convergence curve of the UAV fitness value, with the red circle indicating the differential seeking advantage and the blue asterisk indicating the adaptive seeking advantage. We use the method of accurate calculation of the cost of the trajectory, the accurate range cost through the 3D vertical cut, then a series of obstacles such as mountainous terrain type will be used as part of the cost of affecting the range; this method is not the actual shortest range, but makes full use of the 3D terrain information, the method can improve the reliability and reasonableness of the implementation results. Figure 12 reveals that adaptive finding can expand the algorithm search space and find more local advantages. At the same time, FDS-ADEA can converge quickly in a limited time.

**Table 4.** Initial parameters of MUCTP.

|  | 1 | 2 | 3 | 4 | 5 | 6 | 7 | 8 |
|---|---|---|---|---|---|---|---|---|
| $U_{pos}(km)$ | (20,100,15) | (25,55,12) | (50,40,13) | (80,50,15) | (98,74,17) | (130,30,12) | (162,27,12) | (175,70,13) |
| $T_{pos}(km)$ | (30,170,20) | (90,210,13) | (115,180,12) | (130,220,13) | (145,200,23) | (160,165,12) | (175,173,11) | (180,150,12) |
| $R_{pos}(km)$ | (130,70,4,23) | (120,140,4,26) | (70,110,5,30) | (60,180,5,25) | (20,80,80) | (120,120,30) | (50,200,50) | (100,50,50) |
| $D(km)$ | 1300 | 1400 | 1500 | 1500 | 1450 | 1500 | 1400 | 1500 |
| $V(km)$ | (40 50) | (30 60) | (20 30) | (35 45) | (30 50) | (30 60) | (20 30) | (35 45) |
| $T_{time}(s)$ | 20 | 10 | 20 | 15 | 20 | 20 | 15 | 10 |
| $Umissile(Pcs)$ | 2 | 1.5 | 1 | 2 | 1.5 | 1 | 1.5 | 1 |
| $OrdT$ | (3 2) | (5 2) | (8 1) | (7 4) |  |  |  |  |
| $T_{wait}(s)$ | (1 5) | (2 7) | (4 10) | (1 6) | (2 4) | (3 10) | (5 12) | (6 9) |

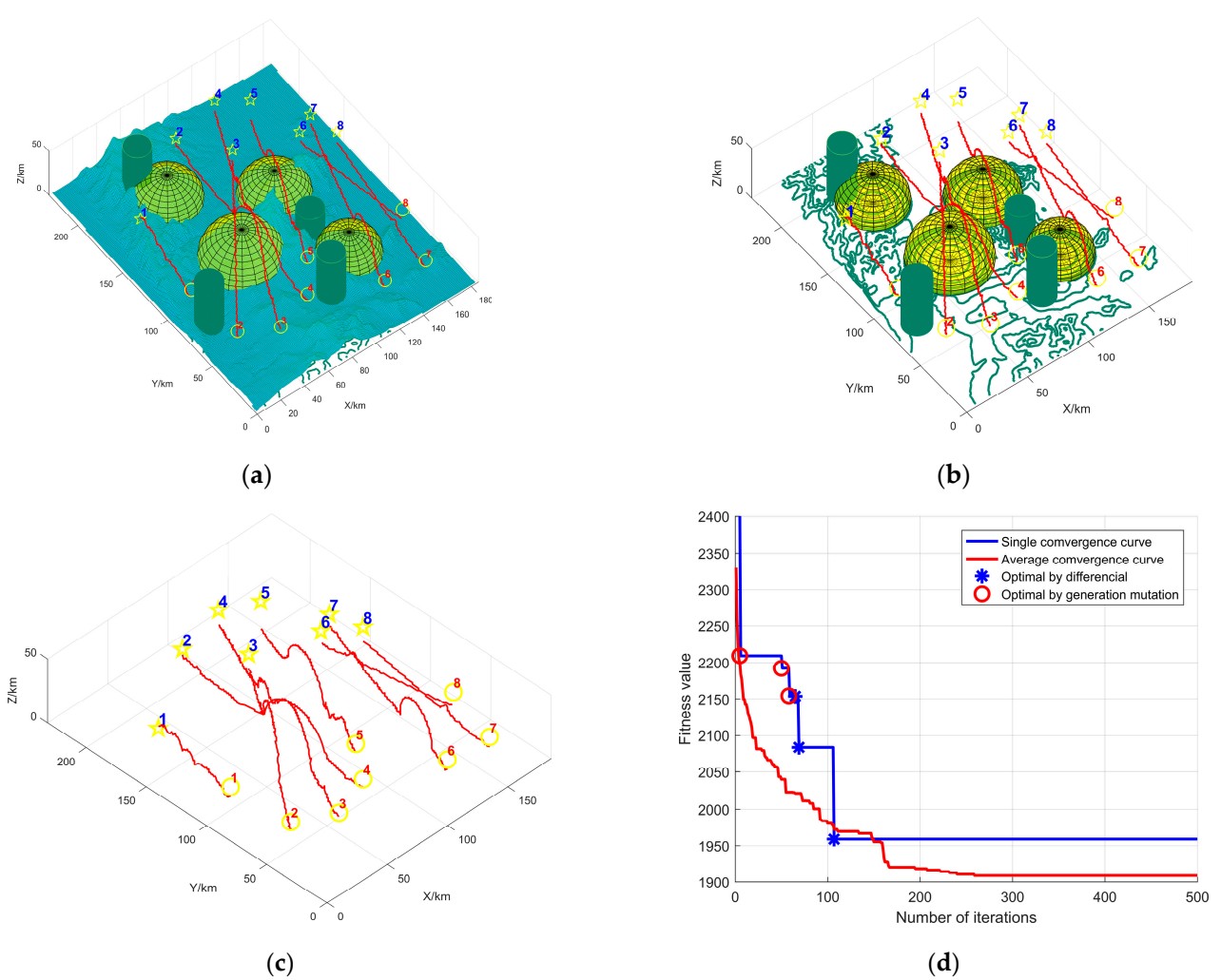

**(a)**        **(b)**

**(c)**        **(d)**

**Figure 12.** Multi-UAV coordinated trajectory execution relationship. (**a**) shows the multi-UAV cooperative trajectory execution relationship; (**b**) hides the mountainous terrain type and shows the effect of each obstacle on the UAVs; (**c**) can visually represent the planning of each UAVs; and (**d**) shows the convergence curve of the total adaptation value of each UAV.

Table 5 shows the MUCTP execution relationship data, where *SolU* and *SolT* represent the sequences of UAVs and mission points, respectively. Besides, *SolC* is the fitness value of each UAV to perform the mission, *SolF* is the total flight cost, and Time is the planning time.

**Table 5.** Cooperative path planning for multiple UAVs.

| Experiment | Parameter | Execute Relationships | | | | | | | | *SolF* (km) | *Time* (s) |
|---|---|---|---|---|---|---|---|---|---|---|---|
| | *SolU* | 1 | 2 | 3 | 4 | 5 | 6 | 7 | 8 | | |
| Experiment 3 | *SolT* | 1 | 3 | 4 | 2 | 5 | 7 | 8 | 6 | 1958.91 | 13.04 |
| | *SolC* | 190.53 | 382.03 | 601.65 | 382.03 | 443.65 | 392.71 | 155.30 | 139.66 | | |

*5.3. Experiment 3: The Influence of Spatial Feasible Domain on the Performance of FDS-ADEA Was Analyzed*

Given the complexity of three-dimensional spaces, the calculation of key trajectory points cannot cover all key points in space. Spatially feasible domain $Pa = [\Delta x, \Delta y, \Delta z]$ quantifies the set of spaces according to actual missions and environmental information and calculates only the key points in $Pa$. However, parameter value $Pa = [\Delta x, \Delta y, \Delta z]$ of the feasible region are determined with some randomness. In this section, four parameter selection options are selected to analyze the effect of parameters on the generation of key trajectory points. The parameters are shown in Table 6.

**Table 6.** Feasible domain parameter settings.

| Mode | Parameters Value *Pa* (km) |
|---|---|
| Parameter I: | $Pa = [10, 10, 5]$ |
| Parameter II: | $Pa = [10, 15, 5]$ |
| Parameter III: | $Pa = [15, 15, 5]$ |
| Parameter IV: | $Pa = [15, 20, 5]$ |

Meanwhile, four evaluation indicators are selected: running time, optimal cost value, average cost value, and maximum cost value comparison. This index can reasonably reflect the interaction of different $Pa$, fitness values, and key trajectory points. The following conclusions can be drawn from Figure 13 and Table 7:

**Table 7.** Influence of feasible domain parameters on fitness values.

| Parameter Settings | | | Populatio-*n* Size | Iterations Number | Repeat Times | Time (s) | Optimal Cost (km) | Average Cost (km) | Maximum Cost (km) |
|---|---|---|---|---|---|---|---|---|---|
| $\Delta x = 10$ | $\Delta y = 10$ | $\Delta z = 5$ | 50 | 500 | 10 | 15 | 4073.39 | 4146.71 | 4214.45 |
| $\Delta x = 10$ | $\Delta y = 15$ | $\Delta z = 5$ | 50 | 500 | 10 | 16 | 4054.66 | 4152.13 | 4179.47 |
| $\Delta x = 15$ | $\Delta y = 15$ | $\Delta z = 5$ | 50 | 500 | 10 | 15 | 4218.28 | 4381.72 | 4419.61 |
| $\Delta x = 15$ | $\Delta y = 20$ | $\Delta z = 5$ | 50 | 500 | 10 | 15 | 4238.02 | 4292.91 | 4625.65 |

Conclusion 1: As shown in Figure 13a, the key trajectory points can be quickly solved under four different $Pa$ parameter settings. To compare the effect of each indicator in the same graph clearly, the running time is multiplied by 230 s. The difference between the average cost of the total flight, the optimal cost, and the highest generation value in a single parameter scheme is small. According to the abovementioned, the larger the parameter setting of $Pa$, the higher the probability that the algorithm solves the combination of poor key trajectory points in the finite search time.

Conclusion 2: As shown in Figure 13b–d, the feasible domain parameter I: the optimal generation value fluctuates, the average generation value and the highest generation value fluctuate more, and the exploratory nature is better; however, the exploitability worsened. The feasible domain parameter IV, the optimal generation value fluctuates more, the average generation value and the highest generation value fluctuate less, and

the exploratory nature is improved, whereas the average generation value and the highest generation value are smoother, indicating that this parameter scheme can achieve a good balance between exploratory and exploitability.

Figure 14 shows the simulation diagram of MUCTP based on FDS-ADEA. In Figure 14, subpanels (a) and (b) represent the MUCTP simulation diagram in the obstacle environment, and the convergence curve of each UAV execution point. Figure 14a shows that the MUCTP based on FDS-ADEA can effectively generate flight trajectories and meet the constraints and cooperative constraints of each UAV. Figure 14b shows that the UAV performs various mission points with fast and strong convergence.

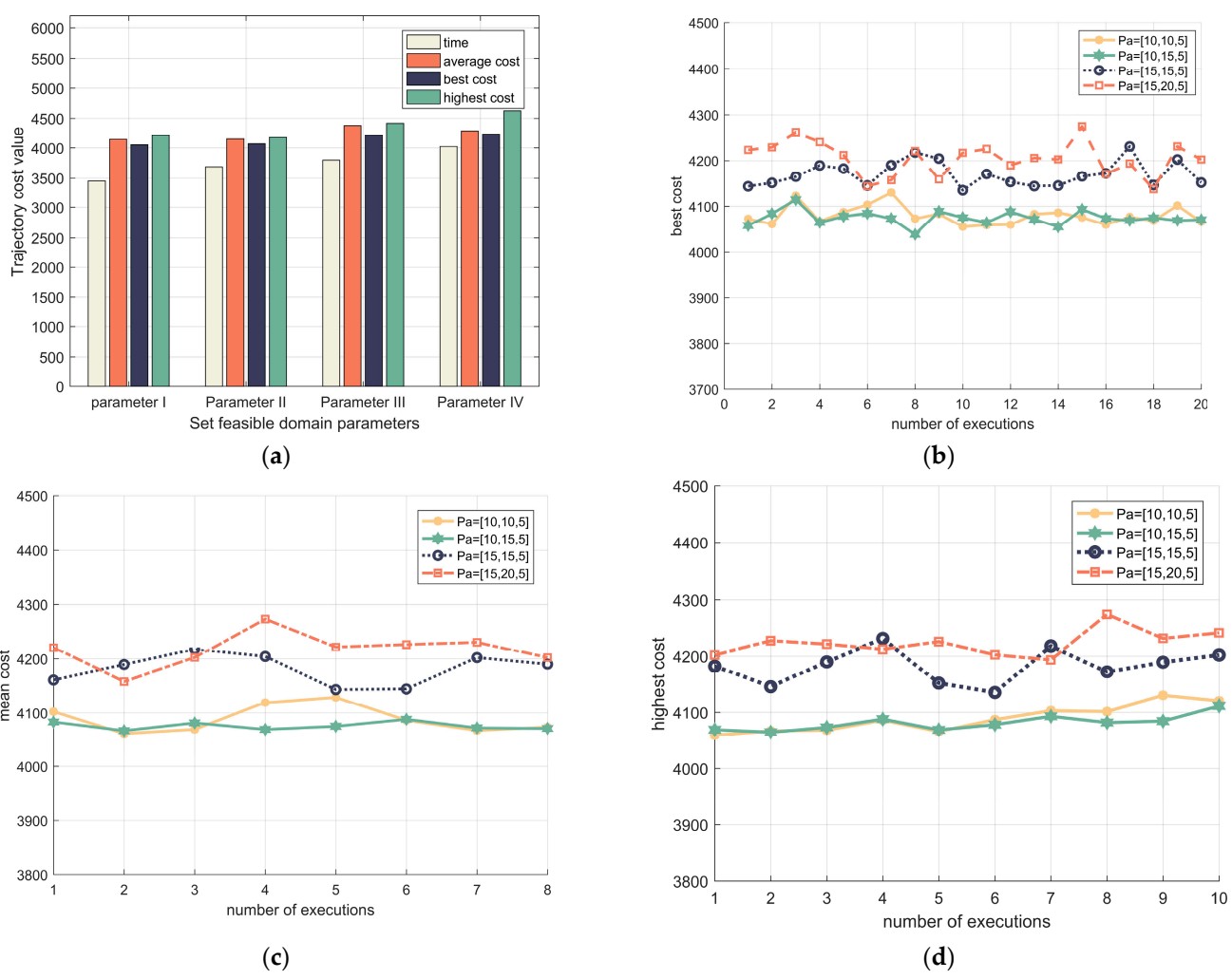

**Figure 13.** Effect of feasible domain parameters on fitness values. (**a**) effect of feasible domain parameters on total fitness values; (**b**) effect of feasible domain parameters on optimal generation value; (**c**) effect of feasible domain parameters on mean algebra value; and (**d**) effect of feasible domain parameters on highest generation value.

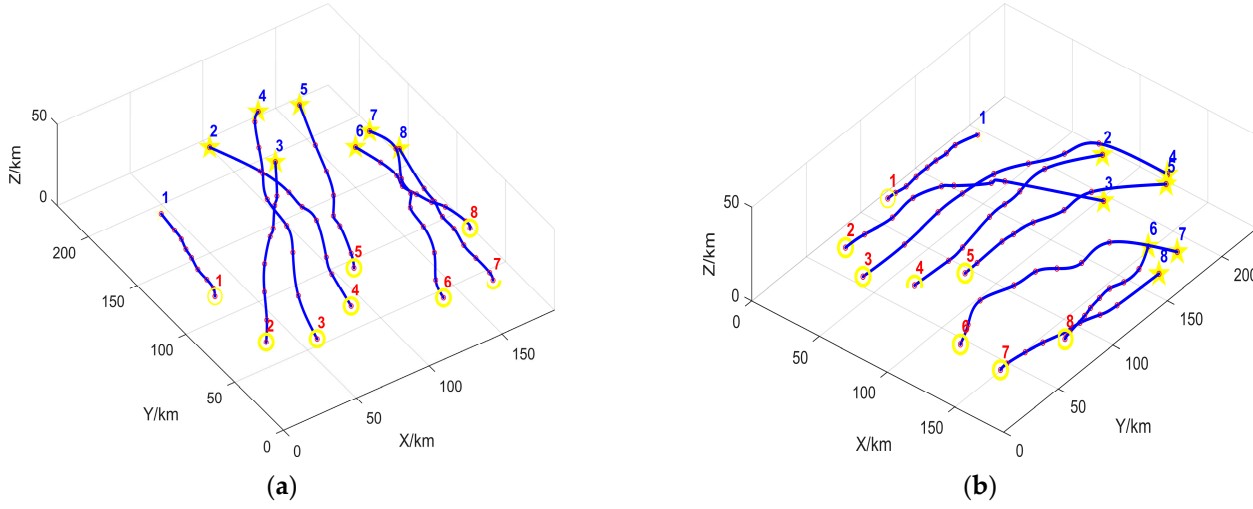

(**a**) MUCTP simulation diagram

(**b**) Convergence curve for each mission

**Figure 14.** Multi-UAV cooperative path planning based on FDS-ADEA.

Figure 15 hides the MUCTP simulation of mountainous and threatening obstacles. The planned trajectory has good cooperative collision avoidance performance, and no intersection exists between the trajectories. In conclusion, FDS-ADEA has good coordination ability.

(**a**)                                                                      (**b**)

**Figure 15.** *Cont.*

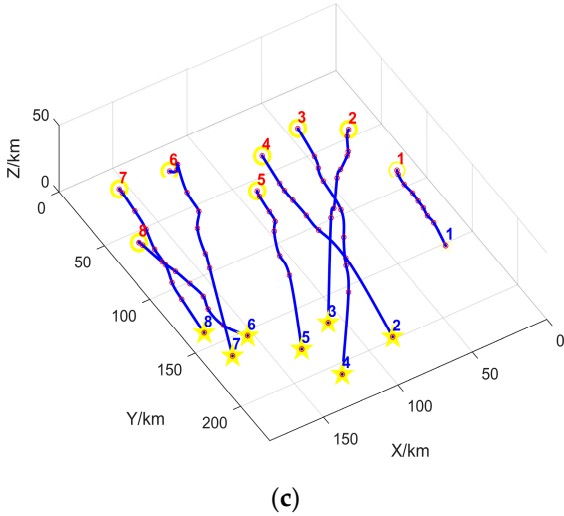

**(c)**

**Figure 15.** Simulation diagram of multi-UAV cooperative trajectory planning; (**a**–**c**) show the multi-UAV cooperative trajectory viewed from different angles.

*5.4. Experiment 4: Comparison of FDS-ADEA and Other Algorithms*

In this section, FDS-ADEA was compared with MSFDE, MOGA, and the FS-ADEA algorithm to verify the improvement of FDS-ADEA's ability to solve key trajectory points. Among them, Multi-Strategy Fusion Differential Evolution (MSFDE) adopts a multi-strategy fusion DEA and the adopted mutation rate $P_c = 0.8$ and crossover rate [2]. Multi-Objective Genetic Algorithm (MOGA) adopts the real-valued cross-variation method, mutation rate $P_c = 0.3$ and crossover rate $S_c = 0.7$ [1]. Full-Scale Adaptive Differential Evolution Algorithm (FS-ADEA) adopts the same algorithm parameters as FDS-ADEA. Other parameters, such as UAVs, number of mission points, population size, and number of iterations are the same. The specific values are shown in Table 8.

**Table 8.** Parameter settings of FDS-ADEA and similar algorithms.

| Experiment | Algorithm | *Num* | $P_n$ | *Gen* | $P_c$ | $S_c$ |
|---|---|---|---|---|---|---|
| Experiment 4 | FDS-ADEA | $N = M = 8$ | 50 | 500 | 0.4 | 0.5 |
| | MSFDE | $N = M = 8$ | 50 | 500 | 0.8 | 0.25 |
| | MOGA | $N = M = 8$ | 50 | 500 | 0.3 | 0.7 |
| | FS-ADEA | $N = M = 8$ | 50 | 500 | 0.4 | 0.5 |

Figure 16 shows the comparison of the average cost value of the four algorithms, indicating that the average cost value of FDS-ADEA is considerably better than that of the other three algorithms. In addition, the average cost is relatively stable, which indicates that FDS-ADEA is more stable than the other three algorithms. The right table records the convergence of the UAV execution mission points of each algorithm. As revealed by data, the average flight cost marked in the gray part is smaller than that of the other three algorithms. Although the average cost value of the adaptive DEA based on key trajectory points is higher than that of the other three algorithms, the difference is small, and the overall cost value is smaller than that of the other three algorithms. Therefore, solving the MUCTP problem on the basis of FDS-ADEA is advantageous.

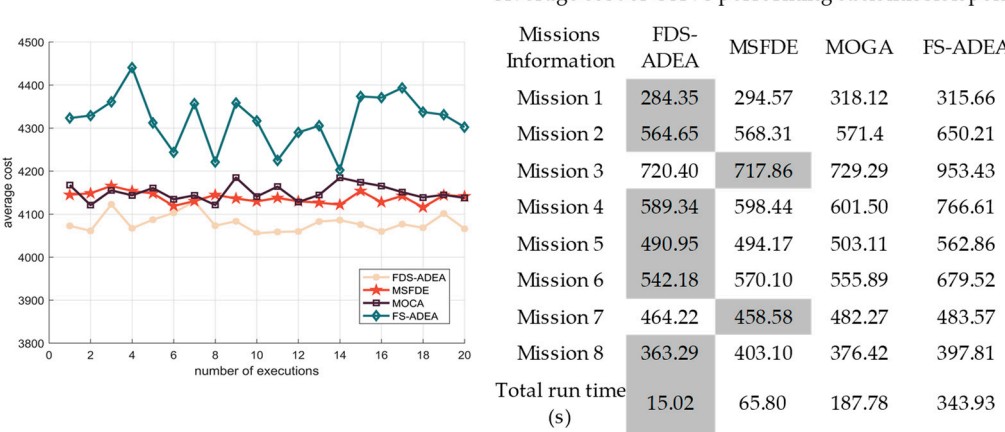

| Average cost of UAVs performing each mission point | | | | |
|---|---|---|---|---|
| Missions Information | FDS-ADEA | MSFDE | MOGA | FS-ADEA |
| Mission 1 | 284.35 | 294.57 | 318.12 | 315.66 |
| Mission 2 | 564.65 | 568.31 | 571.4 | 650.21 |
| Mission 3 | 720.40 | 717.86 | 729.29 | 953.43 |
| Mission 4 | 589.34 | 598.44 | 601.50 | 766.61 |
| Mission 5 | 490.95 | 494.17 | 503.11 | 562.86 |
| Mission 6 | 542.18 | 570.10 | 555.89 | 679.52 |
| Mission 7 | 464.22 | 458.58 | 482.27 | 483.57 |
| Mission 8 | 363.29 | 403.10 | 376.42 | 397.81 |
| Total run time (s) | 15.02 | 65.80 | 187.78 | 343.93 |

**Figure 16.** Average generation value comparison between FDS-ADEA and similar algorithms.

## 6. Conclusions

In this work, an adaptive DEA based on feasible domain space is proposed to solve the difficulties of MUCTP. In terms of spatial structure, a three-dimensional feasible domain is constructed, and the influence of feasible domain parameters on key trajectory points is analyzed in consideration of reducing the complexity search spaces. In terms of node information, environmental information, cooperative information, and mission information are added, and the recognition degree of individuals in three-dimensional spaces is improved to avoid the blindness of algorithm search. In terms of algorithms, the adaptive method and coding correction method are established to share information, such as fitness values, generate key trajectory points, and improve the algorithm search efficiency, which aims to balance individual exploration and exploitation. Finally, a cubic B-spline curve is combined to smooth the flight path to ensure the flying ability of UAVs. Experimental results show that the algorithm has faster convergence speed, stronger coordination ability, and more reasonable trajectory group when dealing with multi-aircraft cooperative trajectory planning.

Future research will focus on considering the cooperative multi-UAV trajectory in dynamic environments. Although the algorithm in this study can effectively deal with the problem of MUCTP in a static environment, in real combat scenarios, the real-time changing environment is often more complex and closer to reality. If the method of trajectory planning in static environment is still followed, the global trajectory will mistakenly guide UAVs into the radar scanning area or fail to reach the specified position, resulting in mission failure. Therefore, MUCTP in dynamic environments is more important than MUCTP in static environments.

## 7. Patents

This section is not mandatory but may be added if there are patents resulting from the work reported in this manuscript.

**Author Contributions:** Conceptualization, G.H. and M.H.; methodology, G.H. and M.H.; project administration, M.H.; software, G.H. and M.H.; supervision, M.H.; validation, X.Y. and P.L.; All authors have read and agreed to the published version of the manuscript.

**Funding:** This research was funded by National Natural Science Foundation of China, grant number 61403416.

**Institutional Review Board Statement:** Not applicable.

**Informed Consent Statement:** Informed consent was obtained from all subjects involved in the study.

**Data Availability Statement:** Not applicable.

**Acknowledgments:** Thanks to Min Hu for his important technical help and for providing experimental equipment and related materials.

**Conflicts of Interest:** The authors declare no conflict of interest.

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
