# Peer review of "Multi-UAV Cooperative Trajectory Planning Based on FDS-ADEA in Complex Environments"

_drones, doi:10.3390/drones7010055_

Round 1

Reviewer 1 Report

1. Lack of a processing flow diagram of the whole algorithm of the paper. It is better to express clearly the connection of each part  such as environment perception, obstacle extraction, path selection and trajectory optimization.It is suggested to provide an algorithm flow diagram more conducive to readers quickly understand the work of the article.

2.The title is aimed at complex environment, but the environment involved in the paper is still very traditional environment, which does not reflect the complexity. Please further explain where the environmental complexity is reflected?

3.In the  engineering application, it is better to explain that the calculation of this paper is the distributed computing or central computing.

4. Line 21 of the abstract should be a b-spline.

5. In this paper, "key trajectory points, candidate key trajectory points, and candidate trajectory points, candidate critical trajectory points ", shoud be  consistent.

  • 6. Line 335 should be Figure 6

Author Response

Dear Reviewer,

Thank you very much for your valuable comments on our manuscript “Multi-UAV cooperative trajectory planning based on FDS-ADEA in complex environments”. Those comments are all valuable and very helpful for revising and improving our manuscript, and significantly guide our research. Following your comments, we have conducted the following revisions with the revised contents being marked in Red, after a deliberate investigation and consideration.

Reviewer 2 Report

The article is of interest to route planning researchers.

Notes:

It is recommended to show formally the stability of the proposed method.

It is recommended to add a full-scale experiment in addition to the computational one.

Information on working with dynamic obstacles should be added.

Information should be added on how to work in case of conflict situations in difficult terrain.

Author Response

(The authors gave the same response as above.)

Round 2

Reviewer 1 Report

  •  
  • The flight experimental studies should be strengthened later.